# Electrosynthesis of 1,4-bis(diphenylphosphanyl) tetrasulfide via sulfur radical addition as cathode material for rechargeable lithium battery

Dan-Yang Wang[1,2], Yubing Si[1,2], Wei Guo[1] & Yongzhu Fu [1✉]

Organic electrodes are promising as next generation energy storage materials originating from their enormous chemical diversity and electrochemical specificity. Although organic synthesis methods have been extended to a broad range, facile and selective methods are still needed to expose the corners of chemical space. Herein, we report the organopolysulfide, 1,4-bis(diphenylphosphanyl)tetrasulfide, which is synthesized by electrochemical oxidation of diphenyl dithiophosphinic acid featuring the cleavage of a P–S single bond and a sulfur radical addition reaction. Density functional theory proves that the external electric field triggers the intramolecular rearrangement of diphenyl dithiophosphinic acid through dehydrogenation and sulfur migration along the P–S bond axis. Impressively, the Li/bis(diphenylphosphanyl) tetrasulfide cell exhibits the high discharge voltage of 2.9 V and stable cycling performance of 500 cycles with the capacity retention of 74.8%. Detailed characterizations confirm the reversible lithiation/delithiation process. This work demonstrates that electrochemical synthesis offers the approach for the preparation of advanced functional materials.

[1] College of Chemistry, Zhengzhou University, Zhengzhou, P. R. China. [2]These authors contributed equally: Dan-Yang Wang, Yubing Si. ✉email: yfu@zzu.edu.cn

Owing to their abundant resources, tunable structures, and environmental benignity, organic compounds as electrode materials for rechargeable batteries have attracted extensive attention[1–4]. Carbonyls, radicals, and organosulfides possess redox-active sites and unique electrochemical properties, affording the potential for next-generation rechargeable lithium batteries[5–9]. For example, cyclohexanehexone ($C_6O_6$) was synthesized and exhibits an ultrahigh capacity of 902 mA h g$^{-1}$ [10]. TEMPO-based catholyte exhibits high energy density for redox flow batteries[11]. However, organic electrodes usually suffer from low electronic conductivity limiting their rate capability, which requires conductive additives[12]. Small organic molecules are soluble in liquid electrolytes resulting in rapid capacity decay in rechargeable metal batteries, which could be improved by polymerization or introduction of polar groups (cyano[13] and sulfonate[14] etc.) or covalently bonding to conductive backbones, e.g., graphene and carbon nanotubes[15]. One important research direction is to develop new organic structures with promising redox activity. For example, the introduction of electron-withdrawing functional groups into the organic structures can generally increase the working voltage of the battery[16–18], organic structures containing multiple redox active sites can have high theoretical specific capacities[19,20]. In addition, scalable synthesis approaches of organic compounds are also needed in order to enable widescale applications.

Among organic electrodes, organodisulfides (R–$S_2$–R, R: organic group) containing S–S bonds were studied firstly by Visco et al.[21,22]. The S–S bonds break during the discharge of lithium batteries, lithium ions, and electrons are ingested and stored[23]. Prior works have been dedicated to identifying the structure–activity relationship of organosulfides as cathode materials. For instance, the synthesized organosulfides from elemental sulfur with vinylic monomers by the "inverse vulcanization" exhibit stable cycling performance[24]. Studies on linear organopolysulfides (R–$S_n$–R, $n > 2$) including dimethyl trisulfide (DMTS) and aromatic polysulfides also reveal their potential for application in rechargeable lithium batteries[25,26]. However, the discharge voltages of organosulfides generally are in the range of 2.0–2.3 V, limiting the specific energy. Thiuram polysulfides and dipyridyl disulfide present higher discharge voltages at 2.6 and 2.45 V, respectively, because of the N-containing heterocycles in those compounds[27–29].

It can be seen that the organic groups bonded with sulfur have a profound effect on the initial discharge voltage of organosulfides. To add new members to the organosulfide family, alternative functional groups with strong electron-withdrawing capability need to be chosen. Phosphorous has a three-valence state, which allows it to be bonded with two phenyl groups and sulfur. Moreover, inorganic phosphate sulfides have shown stable cycling performance in lithium batteries[30,31]. Meanwhile, considering the high conductivity of lithium thiophosphates, organic thiophosphates may exhibit unique property and electrochemical behavior[32,33]. However, few organic thiophosphates containing redox-active sites such as P–S, S–S, and -$S_n$- ($n > 2$) are available. Their electrochemical behavior in lithium batteries is still unknown.

Accordingly, rational design and synthesis of new organopolysulfides are crucial for expanding the organosulfide family and advancing our understanding of their electrochemical behavior in rechargeable lithium batteries. In particular, it is still a great challenge to synthesize organic electrodes with high capacity and high operating voltage. Although researchers have extended traditional organic synthesis methods to a broad range, novel organosulfides compounds, and highly selective methods are still needed to expose the new corners of chemical space[34,35]. Recently, electrochemical synthesis has been adopted widely in the traditional organic reactions, e.g., S–S cross-coupling, C–H functionalization, aryl amination, carbohydroxylation of alkenes, and electrochemical Birch reduction, etc.[36–40]. They can be carried out under mild conditions providing an energy-saving option. In addition, they could have high selectivity and yields. The space for electrochemical synthesis is quite open, which has the potential to enable diverse and functional compounds to be made[41,42]. Diphenylphosphine contains two phenyl groups bonded with a P atom, which is a promising electron-withdrawing group in organosulfides. It has the potential to increase the discharge voltage. In addition, organotetrasulfides have high capacities because of multi-electron involved in the redox reactions. Therefore, 1,4-bis(diphenylphosphanyl)tetrasulfide (BDPPTS) is a promising structure having high discharge voltage as well as high capacity.

In this work, BDPPTS is synthesized by electrochemical oxidation of DPDTP via dehydrogenation and sulfur radical addition reaction (Fig. 1a) and used as a cathode material for a rechargeable lithium battery. This synthesis process is highly selective. When evaluated in lithium half cells, each BDPPTS molecule can take up to 6 Li$^+$ and 6 e$^-$, offering a high theoretical specific capacity of 322.8 mAh g$^{-1}$. Remarkably, it exhibits a high initial discharge voltage at 2.9 V with promising cycling stability. This work provides a viable approach for the synthesis of thiophosphates as electrode materials for rechargeable lithium batteries.

## Results

**Electrochemical synthesis of BDPPTS and reaction process simulation by density functional theory (DFT).** The external electric fields can energize chemical reactions beyond the plain heating and stir as well as laser radiation. The electric field-triggered Diels–Alder reaction, $S_{N1}$ and $S_{N2}$ substitution reactions including cleavage and reformation of C–H, C–O, and O–H bonds are well-explored by experimental and theoretical simulations[43–47]. In this work, the intramolecular rearrangement containing S–H bond activation and sulfur radical addition of DPDTP occurs when an external electric field is applied, leading to the formation of BDPPTS (Fig. 1a). For the electrosynthesis, the pentacoordinate phosphide of DPDTP was firstly dissolved in an undivided cell with lithium bis(trifluoromethanesulfonyl) imide (LiTFSI) and lithium nitrate (LiNO$_3$) in the mixture of 1,3-dioxolane (DOL) and 1,2-dimethoxyethane (DME) as the electrolyte. The cell containing the aforementioned precursor was equipped with a lithium metal plate and carbon paper as the negative and positive electrode, respectively, and charged with Argon gas. After the electrolysis at a constant current of 2 mA for 5 h, the yield of BDPPTS is 96% (Supplementary Fig. 1). Meanwhile, H$_2$ was continuously released as the progress of reaction.

The addition reactions of organic sulfur radical to unsaturated bond are often reported in organic synthesis[48,49]. Although the reports of S radical addition to P=S bond are few, the reaction can occur without extra electric field[50–53]. Under the condition of the external electric field, the dehydrogenation of the thiol group is easy to occur, which could lead to the formation of sulfur radical and the following addition reaction. To improve our understanding of the electrochemical oxidation process, the DFT was employed to investigate the structural characteristics and reaction mechanism. The highest occupied molecular orbital (HOMO) energy level of DPDTP is much higher (more than 1 eV) than those of DOL and DME, DPDTP as the primary electron donor is easily oxidized to release hydrogen (Supplementary Fig. 2). In addition, due to the strong electron-withdrawing effect of diphenylphosphine, the discharge voltage of BDPPTS would be high.

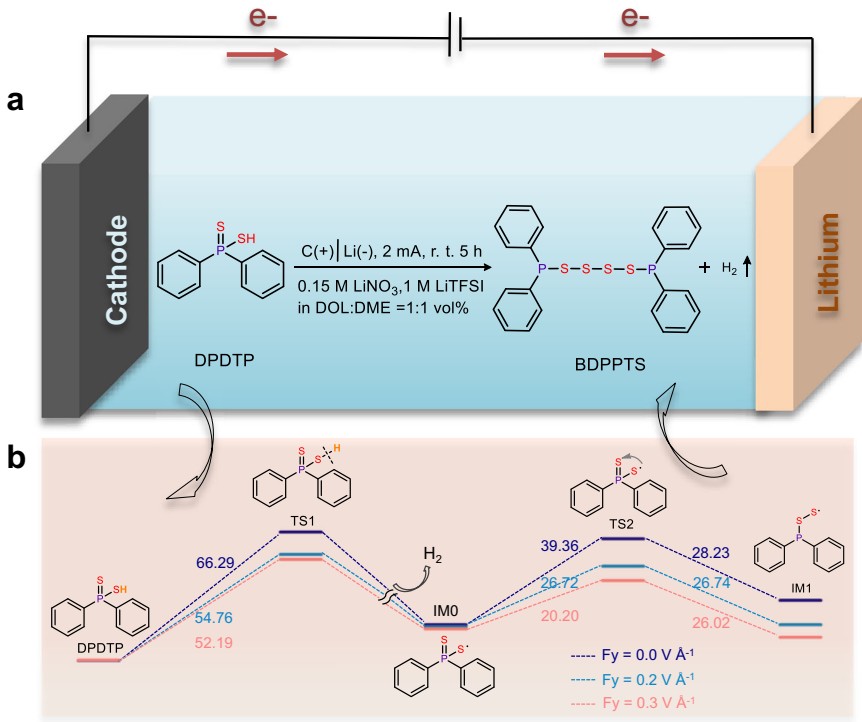

**Fig. 1 Electrochemical synthesis and reaction process simulation of BDPPTS. a** Synthesis of 1,4-bis(diphenylphosphanyl)tetrasulfide (BDPPTS) by electrochemical oxidation of diphenyl dithiophosphinic acid (DPDTP). **b** Proposed reaction process under external electric field: energy profiles (in kcal mol$^{-1}$) for the electrochemical oxidation of DPDTP leading to the formation of linear organopolysulfides BDPPTS (Fy = 0.0 V Å$^{-1}$, dark blue lines; Fy = 0.2 V Å$^{-1}$, blue lines; Fy = 0.3 V Å$^{-1}$, red lines).

As shown in Fig. 1b, when an external electric field is applied along the S–H bond axis (the S–H bond axis is chosen to be the $y$-axis), the disturbance would drive the electron and proton to be separated in different directions, lowering the activation energy of S–H by 12 and 14 kcal mol$^{-1}$ for Fy = 0.2 and 0.3 V Å$^{-1}$, respectively (Supplementary Fig. 3)[54]. Indeed, in the present work, the hydrogen bubble was generated and detected immediately when the electrolysis began. It is worth noting that, in IM0, the thiophosphoryl group (P = S) is just in the right active place and ready for the intramolecular rearrangement, namely, the S atom is prone to migrate along the P–S bond axis then forms the tricoordinate intermediate (IM1) via three-membered transition state, denoted as TS2. Beneficial by the persistency-existing external electric field, the second energy barrier of TS2 decreases about 13 and 19 kcal mol$^{-1}$ at Fy = 0.2 and 0.3 V Å$^{-1}$, respectively, therefore it makes the intramolecular arrangement reaction from IM0 to IM1 thermodynamically and kinetically favorable. Finally, the product of BDPPTS is easily produced with dimerization reaction of IM1s and 39.3 kcal mol$^{-1}$ of energy in total is released.

**Characterization of the synthesized BDPPTS.** Accordingly, in order to confirm the structure of the synthesized compound, Fourier transforms infrared (FTIR) and Raman spectroscopy was employed (Fig. 2a, b). The P = S vibration of DPDTP in the FTIR spectrum appears at 630 cm$^{-1}$, and the characteristic peaks of two P–S bonds are at 430 and 538 cm$^{-1}$. The other peaks are mainly attributed to the composition of the electrolyte (Supplementary Fig. 4). The strong S–H peak of DPDTP can also be detected at 2540 cm$^{-1}$, while the BDPPTS exhibits the disappearance of the peak indicating the departure of H (Supplementary Fig. 5). After the electrochemical oxidation, the P = S vibration peak disappears. Meanwhile, the S–S bond vibration

peak at 420 cm$^{-1}$ appears, affirming the DPDTP conversion is accompanied by a transfer of S atoms and the formation of S–S bonds. To make a further comparison, bis(diphenylphosphinothioyl) disulfide (DPTDS) was synthesized according to the method reported in the literature[55]. It shows both P = S and S–S peaks in the FTIR spectrum (Supplementary Fig. 6), which are in contrast with those of DPDTP. The Raman spectra also show the clear disappearance of P = S bond at 545 cm$^{-1}$, the shift of P–S bond from 645 to 652 cm$^{-1}$, and the strong P–S bond of BDPPTS at 352 cm$^{-1}$. In addition, the characteristic peaks of intramolecular S–S bonds at 425 and 520 cm$^{-1}$ also appear[39,40]. The simulated FTIR and Raman spectra of BDPPTS and DPDTP (Supplementary Fig. 7) are also consistent with the experimental values. The FTIR and Raman analysis reveals the bond transformations during the electrochemical oxidation process, which are consistent with the structural changes from DPDTP to BDPPTS.

Furthermore, $^{31}$P nuclear magnetic resonance (NMR) provides another handle to probe the chemical conversion (Fig. 2c). Clearly, the $^{31}$P chemical shift of DPDTP can be observed at 56.0 ppm. The $^{31}$P chemical shift of BDPPTS appears at 69.0 ppm indicating the change in chemical environment of P atoms. In addition, the $^1$H and $^{13}$C NMR results also confirm the structure of BDPPTS (Supplementary Figs. 8–10). To verify the synthesized BDPPTS, high-resolution mass spectrometry (MS) was performed. The mass spectrum in Fig. 2d shows the positive ion peak at the $m/z$ of 498.9989, which is completely consistent with the molar weight of BDPPTS with a proton. In addition, we also tried to use aliphatic diethyl dithiophosphate (DEDTP) as a precursor to synthesize organopolysulfide. Only the product of bis(diethyl dithiophosphate) BDEDTP was obtained, which still retains the P = S bond after the electrolysis. It is confirmed that the dimeric DEDTP was formed based on the FTIR and MS analysis (Supplementary Figs. 11 and 12). The difference indicates the

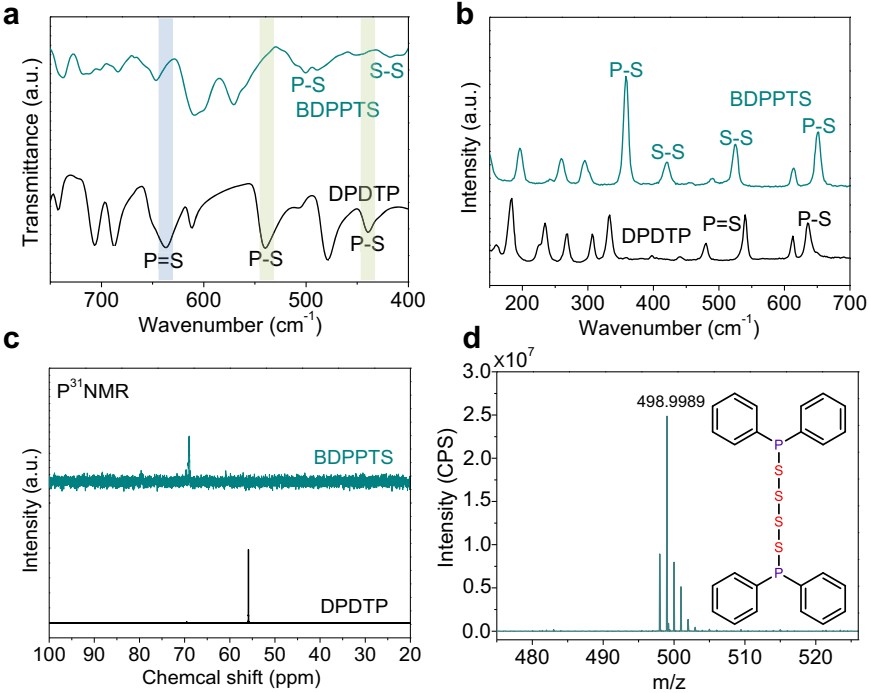

**Fig. 2 Chemical characterization of BDPPTS. a** FTIR spectra. **b** Raman spectra. **c** [31]P NMR spectra of DPDTP and BDPPTS. **d** Mass spectrometry of BDPPTS.

phenyl groups bonded with P atoms can enable sulfur radical addition reaction, whereas the ethyl groups cannot.

**Charge-discharge mechanism of BDPPTS in lithium half cell.** Afterward, BDPPTS was prepared into a catholyte and then added into a carbon nanotube (CNT) paper current collector for the investigation of electrochemical performance in rechargeable lithium battery[56]. This battery test protocol has been widely used in our previous studies with soluble active materials[20,25,57]. The CNTs paper current collector provides efficient electron and ion conduction pathways. More importantly, the soluble BDPPTS and its cycled products can be confined in the nanoscaled space in the CNTs network, minimizing diffusion out of the current collector. In addition, the aromatic groups in BDPPTS could have intermolecular interactions with CNTs via π–π stacking, which further inhibits the shuttle of soluble species upon cycling in the cell[15,58–60]. Scanning electron microscopy (SEM) and energy-dispersive X-ray spectroscopy (EDS) elemental mapping images of the BDPPTS electrode surface are shown in Supplementary Fig. 13. The CNT network structure and distribution of S and P elements on the electrode are clearly expressed. Interestingly, the Li/BDPPTS cell shows a high discharge voltage plateau at ~2.9 V (Fig. 3a), which is beyond those of the inorganic S cathode and conventional organic electrodes (Supplementary Fig. 14). In the following, multiple voltage plateaus are observed. The discharge capacity of BDPPTS is 309 mAh g$^{-1}$, which is 95.7% of the theoretical capacity (322.8 mAh g$^{-1}$) of BDPPTS meaning it can store up to 6 Li$^+$ and 6 e$^-$ per molecule. In addition, the cell can still retain 182.8 mAh g$^{-1}$ at C/10 rate after 100 cycles (Supplementary Fig. 15). Cyclic voltammogram (CV) shows the redox characteristics of the Li/BDPPTS cell (Supplementary Fig. 16). In the cathodic scan, the reduction peaks at 2.8, 2.3, and 2.1 V are observed.

To unravel the structural transformations of BDPPTS during redox in lithium battery, ultra-performance liquid chromatography coupled to quadrupole time-of-flight mass spectrometer (UPLC-QTof-MS) with APCI positive mode was performed

(Fig. 3b). For the discharged product of BDPPTS, peak a at 1.48 min represents the ionized lithium diphenylphosphinothioite (LiDPPT) corresponding to the m/z of 217.0242 (Fig. 3c). Peak b is the incomplete discharged product of BDPPTS confirmed by the m/z of 499.0020 in Fig. 3c, which is very small. Symbol of * represents the electrolyte (Supplementary Fig. 17). After recharge, peak b becomes very obvious, whereas peak a almost disappears, meaning the reversible conversion between BDPPTS and LiDPPT. Interestingly, the low-intensity peak c belongs to 1,5-bis(diphenylphosphanyl)pentasulfide (BDPPPS) suggesting a trace amount of S radicals was added into the structure of BDPPTS forming a longer polysulfide (Supplementary Fig. 18). In addition, very small amounts of bis(diphenylphosphanyl)disulfide and bis(diphenylphosphanyl)trisulfide were found in the charged products by MS (Supplementary Fig. 19).

To further confirm the reversibility of BDPPTS in rechargeable lithium batteries, FTIR of the cycled products after one cycle and mass spectra of the cycled products after 10 and 50 cycles were measured. As shown in Supplementary Fig. 20, the main characteristic band of S–S bond at 430 cm$^{-1}$ is present in the recharged product, while it becomes weak after discharge. The other peaks in the recharged product are consistent with those of BDPPTS except the slight shift of P–S vibration after discharge. In addition, Supplementary Fig. 21 shows the mass spectra of the discharged and recharged products after 10 and 50 cycles. It can be seen that the discharge products after many cycles are the same as those of LiDPPT. While the recharged products mainly are BDPPTS. However, BDPPPS is not found, indicating the BDPPTS is a more reversible recharged product in the lithium cell. Furthermore, [31]P NMR was also measured to investigate the conversion mechanism of BDPPTS (Fig. 3d). An upfield-shifted signal compared to that of BDPPTS appears at 62.5 ppm corresponding to the discharged product LiDPPT. The recharged [31]P NMR spectrum reveals two peaks at 69.5 and 70.5 ppm, which are ascribed to the peaks of b and c in Fig. 3b, respectively. These results are consistent with the UPLC-MS analysis.

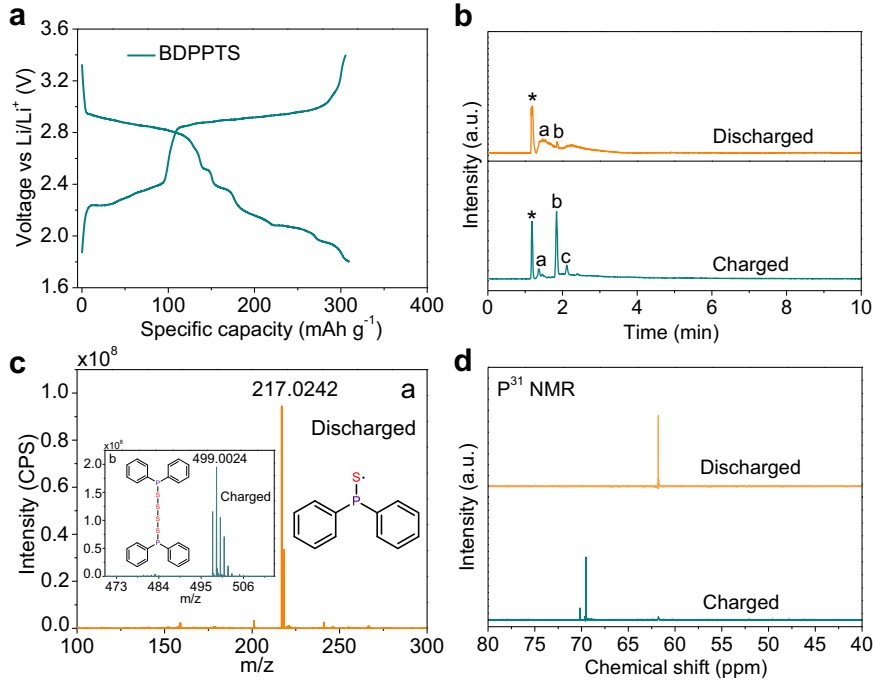

**Fig. 3 Characterization of the cycled products of BDPPTS in a lithium half cell. a** Voltage–capacity profile of a Li/BDPPTS cell in the first cycle at C/10 rate. **b** TICs (total ion current) of BDPPTS, discharged product, and recharged product. **c** m/z of the peak a of the discharged product in (**b**), inset: m/z of the peak b of the recharged product. **d** $^{31}$P NMR spectra of the discharged and recharged products.

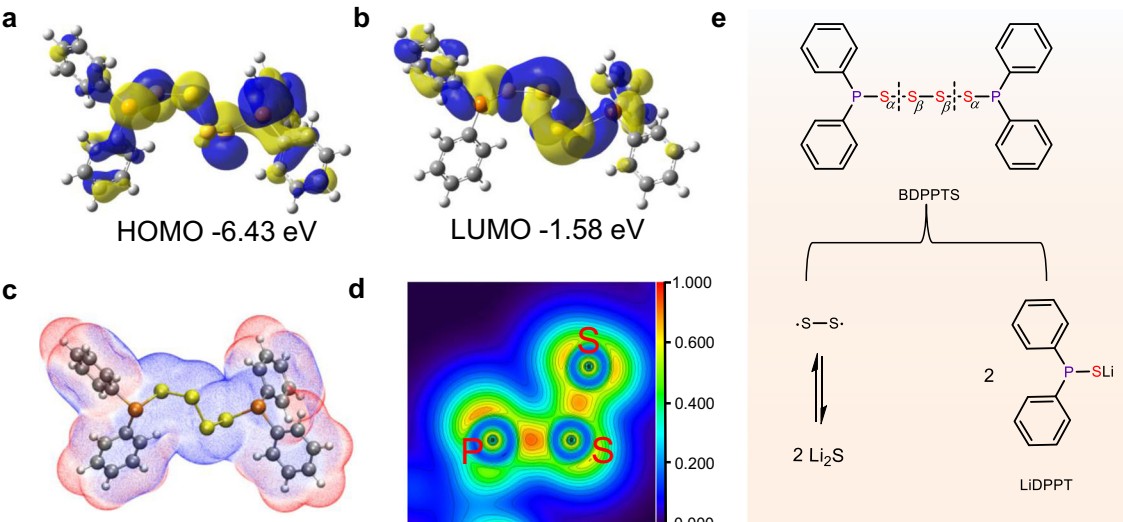

**Fig. 4 Structure simulation and proposed lithiation/delithiation process of BDPPTS. a** and **b** are the HOMO and LUMO plots of BDTTPS, respectively. **c** The isocontour surfaces of ESP for BDPPTS (Gray is carbon, yellow is sulfur, and orange is phosphorus). **d** Localized orbital locator (LOL) map of P–S and S–S bonds of BDPPTS. **e** Proposed electrochemical redox mechanism of BDPPTS in the Li half-cell.

To better understand the redox mechanism of BDPPTS, its HOMO and lowest unoccupied molecular orbital were calculated, as shown in Fig. 4a, b. According to the electrostatic potential of BDPPTS shown in Fig. 4c and Supplementary Fig. 22, the S–S bonds would break first since the reactive sites for electrophilic addition mostly spread along the sulfur chain. Meanwhile, the phenyl groups delocalize the electron density around P atoms in BDPPTS, and thus lower the probability of P–S bond breaking during discharge. The character of the localized molecular orbital for BDPPTS can be seen in Fig. 4d and Supplementary Fig. 23. Based on the above results, the redox mechanism of BDPPTS is elaborated in Fig. 4e. In the discharge process, BDPPTS is

lithiated to form LiDPPT accompanied by the Li-ions attacking the $S_\alpha$ atoms at 2.9 V. This pathway was confirmed by testing the discharged product at 2.7 V. The mass spectrum shows the m/z of 217.0235 correspondings to the ionized LiDPPT and cleavage of $S_\alpha$–$S_\beta$ bond (Supplementary Fig. 24). Afterward, the cleavage of both $S_\alpha$–$S_\beta$ bonds lead to the formation of ·S–S· radical. Further reduction resembles the discharge of elemental sulfur leading to multiple voltage plateaus. At the end of discharge, $Li_2S$ is formed evidenced by the discharge voltage plateau at 2.1 V. The redox process is reversible, $Li_2S$ is oxidized to form S· and ·S–S· radicals which are bonded with DPPT, thus BDPPTS and BDPPPS are formed.

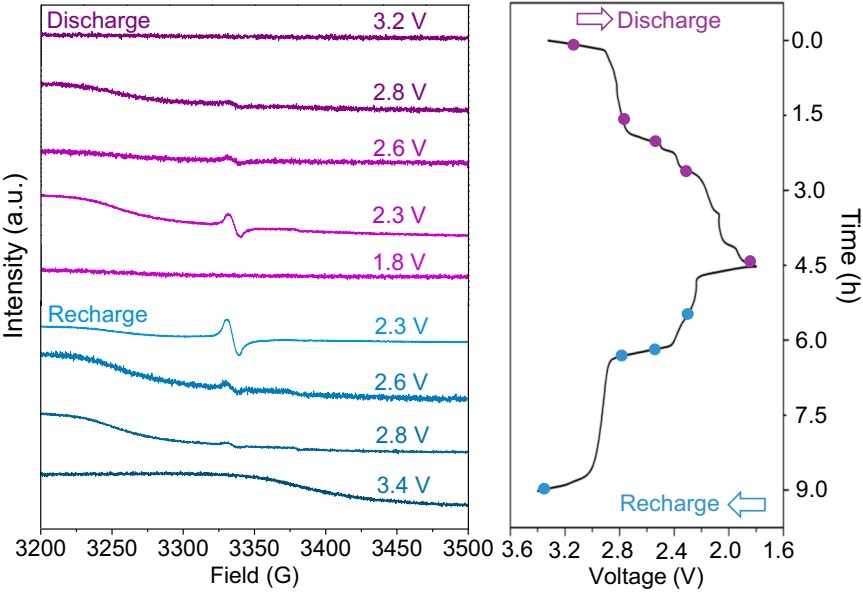

**Fig. 5 Confirmation of sulfur radicals during charge and discharge.** Ex-situ EPR spectra of the BDPPTS electrode during the first cycle of the Li/BDPPTS cell.

To confirm the existence of sulfur radicals, ex situ electron paramagnetic resonance (EPR) was performed on the cycled BDPPTS electrode. It is known that $S_3^{*-}$ radicals are ubiquitous in the redox process of Li–S batteries[61,62], generating from $Li_2S_8$, $Li_2S_6$, $Li_2S_4$, and $Li_2S_2$, but not from $Li_2S$. As shown in Fig. 5, the signal of sulfur radicals appears when the cell was discharged to 2.8 V. After the $S_\alpha$–$S_\beta$ bonds in BDPPTS break, $\cdot S$–$S\cdot$ is formed which would lead to $S_3^{*-}$ radicals. In the following discharge, more $S_3^{*-}$ radicals appear because $Li_2S_x$ is formed. The signal of $S_3^{*-}$ radicals becomes the highest when the cell was discharged to 2.3 V. When the cell was completely discharged at 1.8 V, no $S_3^{*-}$ radicals can be detected as almost all sulfur radicals are converted to $Li_2S$. In the recharge process, the concentration of $S_3^{*-}$ radical increases gradually accompanied by the conversion of $Li_2S$ to $Li_2S_x$ in the voltage range of 1.8–2.3 V. With the consumption of $Li_2S_x$ in the higher charge voltage, a sharp drop of radical concentration is seen due to the formation of BDPPTS at end of recharge. Accordingly, the EPR analysis is consistent with the charge and discharge mechanism proposed above. In addition, to identify the discharge product of BDPPTS, X-ray photoelectron spectroscopy (XPS) analysis was performed. Clearly, the S $2p$ spectrum shown in Supplementary Fig. 25 can be deconvoluted into three pairs of doublet peaks. The doublet peaks of S $2p_{3/2}$ and S $2p_{1/2}$ centered at 160.2 and 161.3 eV, respectively, are attributed to $Li_2S$. The peaks located at 161.9 and 163.0 eV are assigned to $Li_2S_2$ as an incomplete discharged product. The doublet peaks at 163.2 and 164.3 eV are assigned to LiDPPT. Therefore, the XPS results confirm that the discharged products of BDPPTS mainly are LiDPPT and $Li_2S$, which are also consistent with the proposed redox mechanism of BDPPTS.

**Cycling performance of BDPPTS.** Subsequently, the long-term cycling performance of BDPPTS was evaluated at C/5 rate. The Li/BDPPTS cell exhibits the initial specific capacity of 311.6 mAh g$^{-1}$, ending up with 57% retention of the initial capacity after 200 cycles (Fig. 6a). The Coulombic efficiencies mostly are above 98.8%. The shuttle effect is not severe, which further proves the withholding capability of the CNTs current collector on soluble active materials. SEM and EDS mapping on the cycled lithium metal anode and cathode at C/24 rate were also measured to reflect the shuttle of

BDPPTS (Supplementary Figs. 26 and 27). It can be seen that only a small amount of active substances exist on the lithium metal surface after cycling, while most active material still exists on the cathode side. Figure 6b shows the selected charge–discharge voltage profiles. It can be seen that the high voltage plateaus are shorted over cycles and the low voltage regions shrink more obviously. It may be due to the loss of $Li_2S_x$ resulting in capacity decay. The large polarization lies in the region of sulfur-like redox process within the voltage window of 1.8–2.6 V. It is well-known that sulfur undergoes multiple redox processes involving breaking/reformation of S-S bonds resulting in large polarization. In particular, the formation of $Li_2S$ results in the discharge voltage plateau at 2.1 V, and the activation of $Li_2S$ in the recharge is difficult[63]. Furthermore, the electrolyte amount was also optimized. Supplementary Fig. 28 presents the cycling performance of the cells with BDPPTS mass loading of 2.9 and 4.5 mg cm$^{-2}$ with the reduced electrolyte/BDPPTS ratio of 7.8:1 and 5:1 mL g$^{-1}$, respectively. The initial discharge specific capacities are 210.2 and 183.8 mAh g$^{-1}$, respectively. After 60 cycles, the capacity retentions are 81.4 and 79.9%, respectively, and the Coulombic efficiency is above 98% for all cycles.

To demonstrate our hypothesis, a Li/BDPPTS cell at a higher cut-off voltage at 2.5 V was evaluated. Figure 6c shows the cycling performance of the Li/BDPPTS cell. The initial capacity is 89.9 mAh g$^{-1}$ corresponding to transfer of two electrons per BDPPTS molecule. After 500 cycles, the capacity retention is 74.8%. In addition, no obvious increasement in overpotential upon cycling is observed. The voltage profile of the cell shown in Fig. 6d only reveals the reversible redox process of the high voltage plateau at 2.9 V. The sulfur-like voltage profile disappears because the intermediate sulfur species in BDPPTS are not converted to $Li_2S$. In this cut-off voltage range, the cycling stability is improved although the capacity is reduced. The mass spectra of the charged and discharged products after 10 and 50 cycles were also measured (Supplementary Fig. 29). No matter after 10 or 50 cycles, the discharged product is confirmed to be LiDPPT and the charged product is BDPPTS. We believe that both $S_\alpha$–$S_\beta$ bonds in BDPPTS break simultaneously and the intermediate sulfur species in the form of radicals remain in the cell, which lead to the sulfur-like voltage plateaus when discharged to 1.8 V. In the recharge, they return to the structures of BDPPTS (mainly), as shown in the proposed mechanism in Fig. 4e.

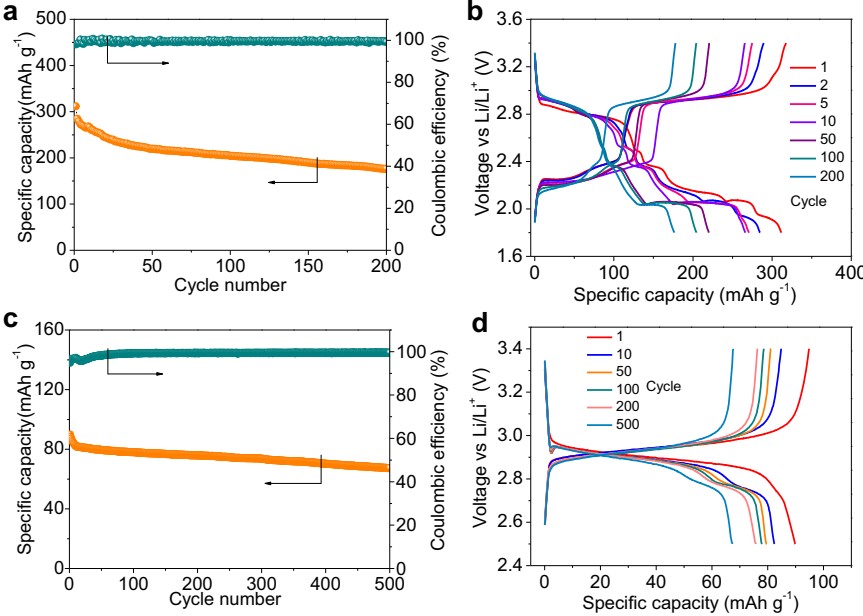

**Fig. 6 Cycling performance of the Li/BDPPTS cell. a** At C/5 rate with the cut-off voltage between 1.8 and 3.4 V and **b** selected charge–discharge voltage profiles. **c** At C/5 rate with the cut-off voltage between 2.5 and 3.4 V and **d** selected charge–discharge voltage profiles.

## Discussion

In summary, we report the account of utilizing electrochemical oxidation to synthesize a organopolysulfide containing phosphorus heteroatoms that can be a promising cathode material for rechargeable lithium batteries. During electrochemical oxidation, the electric field promotes the departure of hydrogen, the cleavage of P–S single bond, and the addition reaction of S radical to P = S bond leading to the formation of BDPPTS. This electrochemical synthesis process is highly selective and the yield can be as high as 96%. Importantly, when it is used as a cathode material, the Li/BDPPTS cell exhibits a high discharge voltage at approximately 2.9 V, which exceeds those of elemental sulfur and most organic electrodes. Utilizing the extensive chemical and structural characterization, the redox process of BDPPTS is revealed. The cleavage of $S_\alpha$–$S_\beta$ and $S_\beta$–$S_\beta$ bonds result in the formation of LiDPPT and $Li_2S$ during discharge. Upon recharge, the addition of sulfur radicals to DPPT yields BDPPTS and BDPPPS. The Li/BDPPTS cell presents stable cycling performance with a capacity retention of 74.8% in the voltage between 2.5 and 3.4 V after 500 cycles. BDPPTS does not have ultrahigh capacity because of the high molecular weight of the organic groups. The role of $Ph_2P$ groups is to increase the initial discharge voltage to 2.9 V thus leading to the increment of specific energy. To improve the practicality of BDPPTS, the cycling stability in the voltage window of 1.8–3.4 V needs to be improved, so that its high specific energy can be maintained. Strategies such as using polysulfide hosting materials or adsorption materials need to be adapted in the future. After all, this facile synthetic approach and intriguing redox process of BDPPTS would motivate more interest in the synthesis and application of organopolysulfides as advanced functional materials.

## Methods

**Electrochemical synthesis of BDPPTS**. In an undivided three-necked bottle (25 mL) equipped with a stir bar, 0.15 mmol DPDTP and 3 mL electrolyte were added and mixed. The bottle was equipped with a piece of lithium metal plate with a surface area of 1.91 cm² and carbon paper with 1.13 cm² as the negative and positive electrode, respectively. Then the electrolysis system was stirred at a constant current of 2 mA and room temperature under Argon gas for 5 h. The chemicals details are described in the "Supplementary methods".

**Synthesis of bis(diphenylphosphinothioyl) disulfide (DPTDS)**. The three-necked and round-bottomed flask equipped with a thermometer, dropping funnel, and argon inlet was charged with DPDTP (0.5 mmol) and 5 mL methyl-*tert*-butyl ether (MTBE). The system was maintained to 2–5 °C in an ice bath. Then the aqueous hydrogen peroxide (0.3 mmol) was dropped into the solution within 15 min, accompanied by the precipitation of white solids. The mixture was stirred at 5 °C for one hour and then filtered. The residue was washed by the MTBE and dried in a vacuum oven at 50 °C. The white solid and 80% yield were obtained.

**Preparation of CNT current collector**. Totally, 160 mg of CNTs (Nanostructure and Amorphous Materials, Inc.) were dispersed in a miscible solution of de-ionized water (500 mL) and isopropyl alcohol (20 mL) by ultrasonication for 15 min, followed by vacuum filtration to render a free-standing CNT paper. The CNT paper was dried in an air oven for 12 h at 100 °C before being peeled off and punched out into circular disks with a diameter of 1.2 cm.

**Preparation of catholyte, cell fabrication, and electrochemical evaluation**. The catholyte was prepared by dissolving BDPPTS in the electrolyte (1.0 M LiTFSI/0.15 M LiNO3 in a mixture of DME/DOL, 1:1 v/v) to render 0.15 M solution in an Argon-filled glove box. Coin cells CR2032 were fabricated in the glove box. First, 20 μL of BDPPTS catholyte was added into the CNT paper current collector, thus the mass loading is approximately 1.5 mg. A Celgard 2400 separator was placed on the top of the CNT paper electrode followed by adding 20 μL of electrolyte on the top of the separator. Subsequently, a lithium metal anode was placed on the separator. The cell was crimped and taken out of the glove box for testing. This cell configuration has been demonstrated to be effective in evaluating soluble active material in lithium batteries. The cells were galvanostatically cycled at 1.8–3.4 V on a LAND battery cycler at different C rates (1 C = 322.8 mA g⁻¹, based on the mass of BDPPTS in the cells). One cell was cycled at 2.5–3.4 V for 500 cycles.

**Characterizations**. The UPLC-QTof-MS was performed on Waters Xevo G2-XS QTof Acquity. The samples were tested by using positive APCI ion source mode. ¹H NMR, ¹³C NMR, and ³¹P NMR spectra were recorded on 600 MHz Bruker spectrometers (600 MHz for ¹H NMR, 150 MHz for ¹³C NMR, 262 MHz for ³¹P NMR) in CDCl3 with TMS as internal standard. Chemical shifts are relative to the chemical shift of CDCl3 with 7.26 ppm in ¹H NMR and 77.16 ppm in ¹³C NMR. XPS measurement was performed with a Thermo Scientific K⁻Alpha⁺ spectrometer with monochromatic Al Kα radiation. EPR spectrometry was performed by Bruker (EMXplus-9.5/2.7). The field was swept 1000 G in 50 seconds and modulated at 100 kHz with 15 G amplitude. The CNTs paper electrodes with 1 mg of BDPPTS were tested at different voltage stages during charge and discharge. FTIR spectra were recorded on a Bruker Tensor II infrared spectrometer, which can directly test solid or liquid samples without tablet compression. The sample after electrochemical oxidation was dropped directly on the test platform for the measurement. CV was performed on a BioLogic VSP potentiostat. SEM of the electrodes was performed on a Zeiss Sigma 500 SEM apparatus. The elemental mapping was examined with EDS attached to the SEM. To analyze the cycled

products, CNT paper after discharge/charge was immersed in 2 mL chromatographic acetonitrile and fully shaken. Then 100 μL of the solution was added into a LC–MS vial with 1 mL of additional chromatographic acetonitrile. The chromatographic column is Waters Acquity UPLC BEH C18 with 2.1 × 100 mm, 1.7 μm particle size, which can effectively separate the products after charge and discharge even in the pure chromatographic acetonitrile mobile phase. Therefore, all LC–MS applications were performed under the conditions of pure chromatographic acetonitrile mobile phase. The injection volume was 1 μL and the solvent phase velocity was 0.2 mL min$^{-1}$.

**Computational details**. The molecular geometries were optimized by the DFT approach at the M062X/def2SVP level, the solvent effect was mimic by the implicit solvent models based on the density approach with the static dielectric constant of DOL/DME set to 7.07[64,65]. For the transition states, the geometries were located with only one imagine frequency, the minimum energy path between the intermediates was verified by the intrinsic reaction coordinate algorithms (Supplementary Tables 1 and 2). All the calculations were performed using the Gaussian 16 program[66].

## Data availability

Source data are provided with this paper. The other data that support the findings of this study are available from the corresponding author upon request.

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

## Acknowledgements

This work was supported by the National Natural Science Foundation of China (Grant nos. 21975225, 51902293, and U2004214) and the China Postdoctoral Science Foundation (Grant no. 2020M682329).

## Author contributions

Y.F. and D.-Y.W. conceived and designed the experiments. D.-Y.W. performed the experiments and Y.S. carried out the simulations. D.-Y.W., Y.S., W.G., and Y.F. analyzed and interpreted the data, D.-Y.W., Y.S., W.G., and Y.F. co-wrote the paper. Y.F. supervised the project.

## Competing interests

The authors declare no competing interests.
