## [Peer Review File · Nature Communications]

Reviewer #1 (Remarks to the Author):

The authors report the interesting electrochemical synthesis of an organopolysulfide, 1,4-bis(diphenylphosphanyl)tetrasulfide (BDPPTS), and its electrochemical performances in lithium batteries. The authors are presenting an electrochemical synthesis (claimed to be for the first time) of the 1,4 bis(diphenylphosphanyl)tetrasulfide (BDPPTS) and use it as a new cathode material for a rechargeable lithium battery. Indeed, a synthesis of 1,4 bis(diphenylphosphanyl)tetrasulfide (BDPPTS) is not reported in the literature till now and full characterizations its is necessary. In this idea, the authors use the FTIR, Raman spectroscopy, Mass spectrometry, and $^{31}\text{P}/^{1}\text{H}/^{13}\text{C}$ NMR to analyze the material proving the structure of the synthesized molecule. In addition, to confirm the workable procedure of preparing the organopolysulfide, the aliphatic diethyl dithiophosphate was used. Instead of polysulfide the bis(diethyl dithiophosphate) BDEDTP was obtained, claiming that the phenyl groups bonded with P atoms (in BDPPTS) can enable sulfur radical addition reaction, whereas the ethyl groups cannot. Here, the authors did not mention any reference in order to support the hypotheses or proposed mechanism. As a consequence, the proposed mechanism of formation of the radical species remains unclear and not totally augmented. The authors should provide more literature supporting evidence to explain this.

However, the work lacks of good discussion and justification for their support. There are serious concerns/problems needs to be addressed, and I would not recommend to publish this work in Nature Communication.

There are so many technical issues with his work that I don't even know where to start and how to better present these so I just list them below the way these came up to my mind while reviewing this work.

- Overall manuscript is poorly written, lack a general discussion on organic batteries in the Introduction section and provide fewer justifications for the observations.
- Reaction simulation by DFT was used to justify their claim of the electrochemical synthesis, which is OK! But it would be better if the authors can provide additional references for the proposed synthesis mechanism.
- The reversible lithium battery is not convincing. Once BDPPTS is discharged (reduced), LiDPTP and Li_2S were formed as demonstrated by the authors. Upon charging (reduction), it is not possible to going back. In order to avoid any misleading in the organic battery field, a systematic study of what happens during cycling is needed, otherwise the mechanism stays very ambiguous and unclear.
- The battery test was carried out with dissolved compound in the electrolyte (catholyte), and lithium chips were used as reference and counter electrode. Generally, a dissolved electrode will cause severe shuttle effect, and the compound will be consumed quickly with fast capacity decaying. However, the authors particularly choose this way, so a convincing explanation is very needed here, why the compound was tested this way and how the authors noticed the shuttle effects.
- The authors explain the capacity decaying by "formation of lithium polysulfides which could dissolve in the liquid electrolyte or shuttle to the lithium metal anode resulting in capacity decay". This doesn't make sense because the electrodes are tested in the dissolved phase (catholyte), how the lithium polysulfides further dissolve???

- Related to the redox mechanism of BDPPTS in Fig. 4e, according to the paper see 10.1002/anie.201611691, the S-S-S bond is breaking during the lithiation process, resulting in the formation of Li₂S. If the ionized LiDPPT is formed during the discharge process, on charging there is a possibility to form also biphenyl dithiophosphate, thermodynamically most favorable. Fig 5d shows the cycling performance of the Li/BDPPTS cell in the 2.5-3.5v potential range. The authors report an initial capacity of 89.9 mAh g⁻¹ corresponding to the transfer of one electron per BDPPTS molecule. How the authors can explain the difference of the reached capacity of the same material at the different potential ranges? Maybe different species are formed during cycling within 2.5-3.5V potential window? Additional work should be carried out to figure this out.

- And in general, the reaction mechanism of DPDTP conversion to BDPPTS makes no sense from organic chemistry point of view, and I would recommend better analysis and documentation on what can happen. For example, I might agree with P(=S)S-S(P=S) formation, and maybe subsequent P=S reduction. But the S₄ chain formation no! FTIR and Raman is used to somehow show this but keep in mind that P=S vibration frequency might significantly shift if in ionic form (PS₂⁻) and then in P(=S)S-S(P=S) – as also known for carboxylates vs. carboxylic acid or ester.

- Lines 189-190 “ The Li/BDPPPS cell exhibits the initial specific capacity of 311.6 mAh g⁻¹ 190 ” in the work was involved only Li/BDPPTS not Li/BDPPPS

- Line 216, “ the Li/BDPPPS cell exhibits a high discharge voltage at approximately 2.9 V, which” it is Li/BDPPTS or LiBDPPPS?

- Was there any attempt to perform cyclic voltammetry of BDPPTS, this compound being highly soluble?

- Why only first cycle is shown? What about the later cycles?

- Why the potential window was kept so narrow (1.8 – 3.3 V)?

- Generally, soluble redox species result in large shuttling effect which restricts sharp increase in voltage profiles and also lead to fast self discharge, whereas this case shows sharp increase in voltage plateaus mean the shuttling effect is not severe. How to explain this?

- Protocols for FTIR measurements after electrochemical oxidation has not discussed.

Serious doubts on discharge products:

- Fig. 3c showed the presence of incompletely discharged BDPPTS in noticeable amount. How can still almost full capacity/high coulombic efficiency was obtained?

- It is ambiguous whether LiDPPT is a radical or ionic species. As explained, if DPPT is a radical species, how does the stability is imparted by phenyl rings? Because phenyl rings affect the stability of the radical lying on the α atom with negligible effect on β atom radical.

- Since most of the explanations of the charging and discharging process involve radical species, please support the described mechanism by comprehensive EPR studies. See DOI:

10.1039/c4ee02730b.

- PXRD investigation of cathode materials has become a valuable tool to study in-situ processes of charge-discharge sequences. I would request authors to provide a thorough analysis to prove the reversible occurrence of cell reaction products.

- In line 171, authors claim that “the phenyl groups delocalize the electron density around P atoms, and thus lower the probability of P-S bond breaking”. This looks strange because if P-S bond breaks, the radical would end up on P atom and the resulting diphenyl phosphine radical would be resonance stabilized. The resulting Lithium diphenylphosphide is also a stable species which could be

easily detected ^{31}P NMR.

- In line 180, the existence of Li_2S could be confirmed by PXRD.
- Figure 5b: please provide suitable justification for the large polarization.

To conclude, whereas the work makes part of the ongoing efforts on improving the organic battery science and technology, and on this side I raise no doubt on the importance of such findings and potential science of new chemistries, I find that the technical and scientific claims, results, and analysis are poorly justified. Don't get me wrong, I'm not against publishing in Nat Comm, but as long as the scientific and technical content is bad, I cannot recommend this. SO I would suggest authors to have another closer look at their data, re-analyse, and maybe come up with a better story and explanation.

Reviewer #2 (Remarks to the Author):

The article entitled "Electrosynthesis of 1,4-bis(diphenylphosphanyl) tetrasulfide via sulfur radical addition as cathode material for lithium battery" describes a new organic cathode based on sulfur-sulfur bridges. Generally the work is well done although some points must be clarified. I recommend to consider the paper for publication in Nature Communication after major revision.

Here my comments:

- 1) The electrochemical reaction produce H_2 gas: where does it go? Is cell bubbling during S-S formation? The authors may follow the reaction in a three electrode flask in order to confirm the mechanism. Hydrogen gas may represent a concern for battery safety.
- 2) The authors should consider shuttle effect: what is organic cathode solubility during cycling (especially at low current C/24)? The authors should add SEM image and EDS mapping of lithium metal anode before and after C/24 cycling in order to detect sulfur presence due to shuttle effect and the consequent decomposition.
- 3) Is there any influence of liquid electrolyte on the performances? Is it possible to use carbonate-based electrolyte? Was electrolyte amount optimized (minimized)?
- 4) What happens if Ketjen black carbon is used instead of nanotubes: how do performances change?
- 5) Phosphodithioic acid seems very toxic and it may represent a serious danger if used in battery (as most sulfides and phosphosulfides based electrolyte as well): is it possible to replace it with a chemical less dangerous? In addition: what is its cost? It seems quite expensive for future applications.

Reviewer #3 (Remarks to the Author):

Fu and coworkers describe the synthesis of a novel organopolysulfide, 1,4-bis(diphenylphosphanyl)tetrasulfide (BDPPTS), prepared by facile electrochemical oxidation featuring the cleavage of P-S single bond and sulfur radical addition reaction. The Li/BDPPTS half-cell delivers stable cycling performance of 500 cycles with a high capacity retention of 74.8%. This paper provides a new direction to the synthesis of new organopolysulfides for cathode materials which can advance our understanding of electrochemical behaviors of organic materials in lithium-ion batteries. Further, this system exhibits an impressive discharge voltage plateau, which is beyond

those of the inorganic S cathode and conventional organic electrodes. However, this work lacks the detailed rationality of the dimer (BDPPTS) design, and further explanation of their outstanding electrochemical performance needs to be addressed. Therefore, I cannot recommend publication in Nature Communications until the following issues are carefully considered and addressed:

Comments:

C1: Authors have done a good job including relevant references; however, there are a few more references they should consider including:

1. There are a few examples of organic electrodes system where cycling performance has improved with a limited voltage range, and that should be referenced: Adv. Func. Mater. 2016, 26, 6896-6903
2. I suggest adding some references for the discharge voltage intervals for other systems including carbonyl, radical, and azo systems. These references should be included to highlight the high discharge voltage of BDPPTS system, depicted in Supplementary Figure 11.

C2: Authors use the terms "lithium batteries" and "Lithium-ion batteries" interchangeably. Indeed, these two terms indicate two different things: Lithium batteries are mostly non-rechargeable primary battery with lithium anode. Thus, it is more appropriate to use proper names like "rechargeable batteries" or "lithium-ion batteries" in the text.

C3: There is a repetition of the same sentence, which needs to be removed in Pg 8, line 139-140.

C4: Authors discuss the importance of utilizing highly selective chemical methods to prepare electrode materials. They further argue that these reactions should be conducted under mild condition providing an energy-saving and "greener" option to fabricate organic electrode materials. Although these arguments can be well adopted and justify their approach to designing BDPPTS by electrooxidation reaction, they need to further address the design rationale of this dimer. They emphasize that Li/BDPPTS can discharge at higher voltage, leading to the increased power of the battery; however, there is no explanation on why such a design can lead to the increase in discharge voltage. I suspect that there could be a structural relationship such as phenyl groups (delocalizing electron density) and P atoms in the design have an impact on the redox potential. Authors should comment on why this would be the case.

C5: It is interesting to see that BDPPTS has excellent cycling stability, although it is a small molecular system. Authors claim that the cell configuration using catholyte and CNT have been demonstrated to be useful in evaluating soluble active materials in lithium-ion batteries (In the experimental section). Is this because of intermolecular interaction such as pi-pi stacking? The dissolution in the electrolyte has been a huge hurdle for small molecular systems as organic electrodes. Authors should comment and clarify why they have chosen this specific cell configuration in more details and elaborate on this with more references with similar works on the small molecular system.

C6: One of the advantages of organopolysulfide system is that these can have higher theoretical/specific capacity ($> 500 \text{ mA h g}^{-1}$). Although Li/BDPPTS has a relatively high specific capacity (322 mA h g^{-1}), this number decreases significantly (less than 100 mA h g^{-1}) after the voltage cut-off cycling. Authors should comment on this practical capacity decrease and how does this affect on the practical application for their system.

RESPONSE TO REVIEWERS' COMMENTS

REVIEWER 1:

Overall Comment: *The authors report the interesting electrochemical synthesis of an organopolysulfide, 1,4-bis(diphenylphosphanyl)tetrasulfide (BDPPTS), and its electrochemical performances in lithium batteries. The authors are presenting an electrochemical synthesis (claimed to be for the first time) of the 1,4 bis(diphenylphosphanyl)tetrasulfide (BDPPTS) and use it as a new cathode material for a rechargeable lithium battery. Indeed, a synthesis of 1,4 bis(diphenylphosphanyl)tetrasulfide (BDPPTS) is not reported in the literature till now and full characterizations its is necessary. In this idea, the authors use the FTIR, Raman spectroscopy, Mass spectrometry, and $^{31}\text{P}/^1\text{H}/^{13}\text{C}$ NMR to analyze the material proving the structure of the synthesized molecule. In addition, to confirm the workable procedure of preparing the organopolysulfide, the aliphatic diethyl dithiophosphate was used. Instead of polysulfide the bis(diethyl dithiophosphate) BDEDTP was obtained, claiming that the phenyl groups bonded with P atoms (in BDPPTS) can enable sulfur radical addition reaction, whereas the ethyl groups cannot. Here, the authors did not mention any reference in order to support the hypotheses or proposed mechanism. As a consequence, the proposed mechanism of formation of the radical species remains unclear and not totally augmented. The authors should provide more literature supporting evidence to explain this. However, the work lacks of good discussion and justification for their support. There are serious concerns/problems needs to be addressed, and I would not recommend to publish this work in Nature Communication. There are so many technical issues with his work that I don't even know where to start and how to better present these so I just list them below the way these came up to my mind while reviewing this work.*

Answer to overall comment: We thank the reviewer for the valuable comments. To support the hypothesis/proposed mechanism, we have now added several related literatures (refs. 48-53) and provided additional experimental data including EPR, XPS, mass spectra, and FTIR etc. in the revised manuscript. In addition, we have re-organized our discussion and strengthened our claims. We hope the revised manuscript is now suitable for publication in Nature Communications.

Comment 1: *Overall manuscript is poorly written, lack a general discussion on organic batteries in the Introduction section and provide fewer justifications for the observations.*

Answer to comment 1: We have now added more discussion about organic batteries in the introduction on pages 2 and 3. In addition, we have reduced justifications for the observations in the Introduction section in the revised manuscript.

Comment 2: *Reaction simulation by DFT was used to justify their claim of the electrochemical synthesis, which is OK! But it would be better if the authors can provide additional references for the proposed synthesis mechanism.*

Answer to comment 2: We have now added some relevant references (refs. 48-53) to justify our claim of the electrochemical synthesis. The addition reactions of organic sulfur radical to unsaturated bond are often reported in organic synthesis. Although the reports of S radical

addition to P=S bond are few, the reaction can occur without extra electric field, which has been reported. Under the condition of external electric field, the dehydrogenation of thiol group is easy to occur leading to the formation of sulfur radical and the followed addition reaction. We have now added this point on page 6 in the revised manuscript.

Comment 3: *The reversible lithium battery is not convincing. Once BDPPTS is discharged (reduced), LiDPTP and Li₂S were formed as demonstrated by the authors. Upon charging (reduction), it is not possible to going back. In order to avoid any misleading in the organic battery field, a systematic study of what happens during cycling is needed, otherwise the mechanism stays very ambiguous and unclear.*

Answer to comment 3: The redox reaction of BDPPTS in lithium battery is indeed reversible, which is supported by the experimental data. The recharged product has been examined by UPLC-MS and ³¹P NMR (Figure 3b and 3d), which indicate the mass and chemical shift after cycle are exactly the same as those of the synthesized BDPPTS. In addition, we supplement the FTIR spectra in **Supplementary Fig. 20** to show that the S-S bond peak is strong in the recharged product, while it is weak in the discharge product. This result is also consistent with the redox mechanism, *i.e.*, reversible S-S bond breaking/formation in the discharge/recharge process of BDPPTS in rechargeable lithium battery. Moreover, we have added additional mass spectrometry measurements after 10 and 50 cycles of the Li/BDPPTS cell, which are shown in **Supplementary Fig. 21**. The m/z of the recharged product after multiple cycles is also same as that of BDPPTS, indicating BDPPTS is the main recharged product after many cycles in the lithium cell. Corresponding discussion has been added on page 11 in the revised manuscript.

Comment 4: *The battery test was carried out with dissolved compound in the electrolyte (catholyte), and lithium chips were used as reference and counter electrode. Generally, a dissolved electrode will cause severe shuttle effect, and the compound will be consumed quickly with fast capacity decaying. However, the authors particularly choose this way, so a convincing explanation is very needed here, why the compound was tested this way and how the authors noticed the shuttle effects.*

Answer to comment 4: The battery test protocol used in this work is based on our previous studies on soluble cathode materials, such as lithium polysulfides (*Angew. Chem. Int. Ed.* **2013**, *52*, 6930-6935), dimethyl trisulfide (*Angew. Chem. Int. Ed.* **2016**, *55*, 10027-10031), diphenyl trisulfide (*ACS Energy Lett.* **2016**, *1*, 1221-1226) etc. The binder-free carbon nanotube (CNTs) paper used as current collector provides efficient electron and ion conduction pathways. More importantly, the soluble active materials and their cycled products can be confined in the nanoscaled spaces in the CNTs network, minimizing diffusion out of the current collector. Therefore, their shuttle and loss upon cycling in the battery are suppressed although they are not completely eliminated. This protocol has been successfully used in the previous studies showing high utilization of active materials and quite stable cycle life. Furthermore, the organosulfides with aromaticity appear to be more stable performance, which is also attributed to the intermolecular interaction between benzene ring and carbon nanotubes via π - π stacking. This explanation has now been added on page 9 in the revised manuscript to make it clear.

Comment 5: *The authors explain the capacity decaying by “formation of lithium polysulfides which could dissolve in the liquid electrolyte or shuttle to the lithium metal anode resulting in*

capacity decay”. This doesn’t make sense because the electrodes are tested in the dissolved phase (catholyte), how the lithium polysulfides further dissolve?

Answer to comment 5: We are sorry for the confusing. We have now revised this sentence to “The sulfur-like voltage profile disappears because the intermediate sulfur species in BDPPTS are not converted to Li_2S .” on page 15.

Comment 6: Related to the redox mechanism of BDPPTS in Fig. 4e, according to the paper see 10.1002/anie.201611691, the S-S bond is breaking during the lithiation process, resulting in the formation of Li_2S . If the ionized LiDPPT is formed during the discharge process, on charging there is a possibility to form also biphenyl dithiophosphate, thermodynamically most favorable. Fig 5d shows the cycling performance of the Li/BDPPTS cell in the 2.5-3.5 V potential range. The authors report an initial capacity of 89.9 mAh g^{-1} corresponding to the transfer of one electron per BDPPTS molecule. How the authors can explain the difference of the reached capacity of the same material at the different potential ranges? Maybe different species are formed during cycling within 2.5-3.5V potential window? Additional work should be carried out to figure this out.

Answer to comment 6: We agree with the reviewer’s comments. Indeed, biphenyl dithiophosphate and biphenyl trithiophosphate are formed in the charge process, but in a very small amount. Only in the MS analysis, they are found as shown in **Supplementary Fig. 19**, but the intensities are very low. We have now made an explanation on page 11 to make it clear.

Yes, different discharged products are formed in different potential range. Here, we have to correct the reviewer’s comment, the initial capacity of 89.9 mAh g^{-1} corresponds to the transfer of almost two electrons per BDPPTS molecule, not one. When the battery is cycled within 2.5-3.5 V potential window, the discharged product confirmed by LC-MS is only LiDPPT (Supplementary Fig. 24) and Li_2S is not formed, unlike those of the battery cycled within 1.8-3.5 V potential window. We believe that both $\text{S}_\alpha\text{-S}_\beta$ bonds in BDPPTS break simultaneously and the intermediate sulfur species in the form of radicals remain in the cell, which lead to the sulfur-like voltage plateaus when discharged to 1.8 V. The existence of sulfur radicals is verified by the EPR measurement in Fig. 5. The charged product is still BDPPTS. Even after 10 and 50 cycles, LiDPPT and BDPPTS are still the discharged and charged products that can be confirmed by LC-MS, respectively (**Supplementary Fig. 29**). Additional discussion has been added on pages 15 and 16 in the revised manuscript.

Comment 7: And in general, the reaction mechanism of DPDTP conversion to BDPPTS makes no sense from organic chemistry point of view, and I would recommend better analysis and documentation on what can happen. For example, I might agree with P(=S)S-S(P=S) formation, and maybe subsequent P=S reduction. But the S4 chain formation no! FTIR and Raman is used to somehow show this but keep in mind that P=S vibration frequency might significantly shift if in ionic form (PS^{2-}) and then in P(=S)S-S(P=S)^- as also known for carboxylates vs. carboxylic acid or ester.

Answer to comment 7: Addition of sulfur radical to unsaturated bonds is common in organic synthesis (refs. 48 and 49). In our work, the dehydrogenation of thiol group leads to sulfur radical under electric field, which can readily attack the P=S bond close to it to form S_2 radical, then S_4 chain. In **Supplementary Fig. 4**, we have provided the FTIR spectra of DOL/DME and

electrolytes. Compared with the spectrum of BDPPTS, it can be clearly seen that the P=S peak completely disappears and no close peaks can be assigned to the ionic form (PS^{2-}) or P(=S)S-S(P=S) . To distinguish differences between BDPPTS and compound with P(=S)S-S(P=S) structure, we synthesized bis(diphenylphosphinothio) disulfide (DPTDS, structure is shown below) based on the literature (Org. Process Res. Dev. 2013, 17, 47-52). Its FTIR spectrum is given in **Supplementary Fig. 6**. It can be clearly seen that the P=S band exists in DPTDS, but not in BDPPTS. Furthermore, we also evaluate the DPTDS' performance in lithium battery, as shown in Figure below. It can be seen that its charge and discharge voltage profiles are completely different from those of BDPPTS. BDPPTS exhibits obvious discharge behavior of sulfur, but DPTDS shows two voltage slopes in between there is a discharge voltage plateau at 2.6 V. In summary, BDPPTS is indeed synthesized successfully in our work. We have added this point on page 8.

Figure R1. Discharge/charge voltage profile of DPTDS (left) and its chemical structure (right).

Comment 8: Lines 189-190 “The Li/BDPPPS cell exhibits the initial specific capacity of 311.6 mAh g^{-1} ” in the work was involved only Li/BDPPTS not Li/BDPPPS. Line 216, “the Li/BDPPPS cell exhibits a high discharge voltage at approximately 2.9 V” which it is Li/BDPPTS or LiBDPPPS?

Answer to comment 8: We are sorry for the mistakes. We have now changed them to Li/BDPPTS.

Comment 9: Was there any attempt to perform cyclic voltammetry of BDPPTS, this compound being highly soluble?

Answer to comment 9: We have now provided the cyclic voltammogram of the Li/BDPPTS cell in **Supplementary Fig. 16**. Corresponding discussion has been added on page 9. In addition, the compound is soluble in electrolyte thus it was prepared into the catholyte.

Comment 10: Why only first cycle is shown? What about the later cycles?

Answer to comment 10: The first cycle is shown to present the discharge and charge voltage profile of BDPPTS in rechargeable lithium battery. We have now provided later cycles in **Supplementary Fig. 15**.

Comment 11: Why the potential window was kept so narrow (1.8-3.3 V)?

Answer to comment 11: We have tried higher cutoff voltage of 3.6 V, but no additional oxidation process is revealed. The low cutoff voltage of 1.8 V was kept because LiNO_3 was used in the electrolyte, which could be reduced in battery causing side reactions and irreversible capacities if the cutoff voltage is lower than 1.8 V.

Comment 12: Generally, soluble redox species result in large shuttling effect which restricts sharp increase in voltage profiles and also lead to fast self discharge, whereas this case shows sharp increase in voltage plateaus mean the shuttling effect is not sever. How to explain this?

Answer to comment 12: In our work, LiNO_3 additive was used in the electrolyte. It is widely known that it can passivate lithium metal anode leading to reduced reactions between shuttled lithium polysulfides with lithium metal and significantly improved Coulombic efficiency in Li-S batteries. In this work, LiNO_3 can also protect lithium metal, therefore sharp increase in voltage plateau can be seen and high Coulombic efficiency can be obtained.

Comment 13: Protocols for FTIR measurements after electrochemical oxidation has not discussed.

Answer to comment 13: The FTIR measurements were performed on a Bruke Tensor \square infrared spectrometer, which can directly test solid or liquid sample without pressing tablets. The sample after electrochemical oxidation was dropped directly on the test platform for the measurement. This discussion has been added on page 19 in the revised manuscript.

Comment 14: Fig. 3c showed the presence of incompletely discharged BDPPTS in noticeable amount. How can still almost full capacity/high coulombic efficiency was obtained?

Answer to comment 14: In Figure 3c, the inset figure represents the m/z of the peak b of the recharged product, *i.e.*, BDPPTS, not incomplete discharged BDPPTS. From the discharge plot in Figure 3b, the peak b almost disappears indicating most BDPPTS has been transformed into the discharged produce **a**. This is consistent with the achieved 95.7% of theoretical capacity. In order to make it clear, we have made a new mark on Fig. 3c.

Comment 15: It is ambiguous whether LiDPPT is a radical or ionic species. As explained, if DPPT is a radical species, how does the stability is imparted by phenyl rings? Because phenyl rings affect the stability of the radical lying on the atom with negligible effect on atom radical.

Answer to comment 15: LiDPPT should be an ionic species, not a radical. In the discharge process, DPPT radical is formed and it takes a lithium ion and electron forming LiDPPT. LiDPPT contains a lithium cation and DPPT anion like a salt, which can dissociate in solvent.

Comment 16: *Since most of the explanations of the charging and discharging process involve radical species, please support the described mechanism by comprehensive EPR studies. See DOI: 10.1039/c4ee02730b.*

Answer to comment 16: To confirm the existence of sulfur radicals, ex-situ electron paramagnetic resonance (EPR) was performed on the cycled BDPPTS electrode. It is known that S_3^* radicals are ubiquitous in the redox process of Li-S batteries, generating from Li_2S_8 , Li_2S_6 , Li_2S_4 , and Li_2S_2 , but not from Li_2S . As shown in Fig. 5, the signal of sulfur radicals appears when the cell was discharged to 2.8 V. After the $S_\alpha-S_\beta$ bonds in BDPPTS break, $\cdot S-S\cdot$ is formed which would lead to S_3^* radicals. In the following discharge, more S_3^* radicals appear because Li_2S_x is formed. The signal of S_3^* radicals becomes the highest when the cell was discharged to 2.3 V. When the cell was completely discharged at 1.8 V, no S_3^* radicals can be detected as all sulfur radicals are converted to Li_2S . In the recharge process, the concentration of S_3^* radical increases gradually accompanied by the conversion of Li_2S to Li_2S_x in the voltage range of 1.8-2.3 V. With the consumption of Li_2S_x in the higher charge voltage, sharp drop of radical concentration is seen due to the formation of BDPPTS at end of recharge. Accordingly, the EPR analysis is consistent with the charge and discharge mechanism proposed. The detailed discussion has been added on page 13, and new Figure 5 has been added.

Comment 17: *PXRD investigation of cathode materials has become a valuable tool to study in-situ processes of charge-discharge sequences. I would request authors to provide a thorough analysis to prove the reversible occurrence of cell reaction products.*

Answer to comment 17: We have tried to conduct XRD and *in-situ* XRD measurements during charge and discharge of the Li/BDPPTS cell. However, no crystalline peaks were observed because BDPPTS and the cycled products are all amorphous. Instead of PXRD, we have utilized UPLC-QTOF-MS (**Supplementary Fig. 21**) and FTIR (**Supplementary Fig. 20**) to confirm the cycled products and reversible breakage/formation of the S-S bonds in BDPPTS upon cycling in lithium battery. These results support the redox mechanism of BDPPTS as shown in Figure 4e.

Comment 18: *In line 171, authors claim that “the phenyl groups delocalize the electron density around P atoms, and thus lower the probability of P-S bond breaking”. This looks strange because if P-S bond breaks, the radical would end up on P atom and the resulting diphenyl phosphine radical would be resonance stabilized. The resulting Lithium diphenylphosphide is also a stable species which could be easily detected ^{31}P NMR.*

Answer to comment 18: We have now revised the sentence to “the phenyl groups delocalize the electron density around P atoms in BDPPTS, and thus lower the probability of P-S bond breaking during discharge” on page 12 to make it clear. Yes, lithium diphenylphosphide may be a stable specie. However, only Ph_2PSLi ($LiDPPT$) not lithium diphenylphosphide was detected by ^{31}P NMR and mass spectroscopy, indicating the cleavage of only $S_\alpha-S_\beta$ bonds in BDPPTS during discharge as shown in Fig. 4e.

Comment 19: *In line 180, the existence of Li_2S could be confirmed by PXRD.*

Answer to comment 19: The existence of Li_2S is hardly to be detected by XRD when the formed Li_2S is amorphous, which is a common result of our previous studies on organosulfide

materials. However, we used X-ray photoelectron spectroscopy (XPS) to confirm the existence of Li_2S (**Supplementary Fig. 25**). The doublet peaks of S $2p_{3/2}$ and S $2p_{1/2}$ centered at 160.2 and 161.3 eV, respectively, are attributed to Li_2S . The doublet peaks at 163.2 and 164.3 eV are assigned to LiDPPT. Therefore, the XPS results confirm that the discharged products of BDPPTS mainly are LiDPPT and Li_2S , which are also consistent with the proposed redox mechanism of BDPPTS. We have now added this on pages 13 and 14 in the revised manuscript.

Comment 20: *Figure 5b: please provide suitable justification for the large polarization.*

Answer to comment 20: Figure 5d (now Figure 6d) shows the polarization within the voltage window of 2.6-3.4 V is not large. The large polarization lies in the region of sulfur-like redox process within the voltage window of 1.8-2.6 V. It is well known that sulfur undergoes multiple redox processes involving breaking/reformation of S-S bonds resulting in large polarization. In particular, the formation of Li_2S results in the discharge voltage plateau at 2.1 V and the activation of Li_2S in the recharge is difficult (Ref. 63). Additional discussion has now been added on page 15 in the revised manuscript.

Last Comment: *To conclude, whereas the work makes part of the ongoing efforts on improving the organic battery science and technology, and on this side I raise no doubt on the importance of such findings and potential science of new chemistries, I find that the technical and scientific claims, results, and analysis are poorly justified. Don't get me wrong, I'm not against publishing in Nat Comm, but as long as the scientific and technical content is bad, I cannot recommend this. SO I would suggest authors to have another closer look at their data, re-analyse, and maybe come up with a better story and explanation.*

Answer to last comment: We greatly appreciate the reviewer #1's valuable comments. We have now made significant revisions and added supplementary measurements and data in the revised manuscript to make our claims clear and strong. We hope the changes we have made can clear out all the doubts the reviewer #1 has.

REVIEWER 2:

Overall Comment: *The article entitled "Electrosynthesis of 1,4-bis(diphenylphosphanyl) tetrasulfide via sulfur radical addition as cathode material for lithium battery" describes a new organic cathode based on sulfur-sulfur bridges. Generally the work is well done although some points must be clarified. I recommend to consider the paper for publication in Nature Communication after major revision.*

Answer to overall comment: We thank the reviewer for the positive comments. We have now made significant revisions and added additional data to support our claims. We hope we have now clarified all the points.

Comment 1: *The electrochemical reaction produce H_2 gas: where does it go? Is cell bubbling during S-S formation? The authors may follow the reaction in a three electrode flask in order to confirm the mechanism. Hydrogen gas may represent a concern for battery safety.*

Answer to comment 1: We thank the reviewer for the valuable comments. The electrochemical reaction was carried out in a beaker not in a battery. It does produce H₂ gas, which was released to the atmosphere. We can only notice tiny bubbles during the S-S formation. To confirm the H₂ gas, we used a hydrogen detector close to the reaction. It clearly shows the existence of H₂ gas. After the BDPPTS was synthesized, it was assembled into a testing Li cell. Therefore, no H₂ gas was introduced into the battery.

Figure R2. Photograph of the hydrogen detector showing the detection of H₂ during the electrochemical synthesis.

Comment 2: The authors should consider shuttle effect: what is organic cathode solubility during cycling (especially at low current C/24)? The authors should add SEM image and EDS mapping of lithium metal anode before and after C/24 cycling in order to detect sulfur presence due to shuttle effect and the consequent decomposition.

Answer to comment 2: The organic material is highly soluble in the electrolyte and it has shuttle effect. However, we used binder-free carbon nanotube paper as current collector, which is highly porous and can hold the cycled products within it. In addition, we used LiNO₃ additive in the electrolyte, which can passivate the lithium metal anode preventing chemical reduction of BDPPTS shuttled from the cathode side, leading to high Coulombic efficiency of above 99%. We have now added SEM images and EDS mapping of the lithium metal anode after 5 cycles at C/24 rate in **Supplementary Fig. 26**. Clearly, there is only a small amount of sulfur species on the cycled lithium metal surface. In addition, we have added SEM images and EDS mapping of the cathode after 5 cycles at C/24 rate in **Supplementary Fig. 27**. We can notice that most of active material still exists on the cathode side. These results suggest the shuttle of BDPPTS in the battery is not severe. Additional discussion on this point has been added on page 14.

Comment 3: Is there any influence of liquid electrolyte on the performances? Is it possible to use carbonate-based electrolyte? Was electrolyte amount optimized (minimized)?

Answer to comment 3: Yes, liquid electrolyte has big influence on the performance because BDPPTS is soluble in the electrolyte. Carbonate-based electrolyte is not compatible with BDPPTS because sulfide anions are formed in the discharge which can attack carbonates via nucleophilic reactions. The Li/BDPPTS cell with carbonate electrolyte cannot be cycled. The electrolyte amount we used in our work is optimized. When the electrolyte amount is reduced,

the achievable capacities are low. The cycling performance with reduced electrolyte has now been added in **Supplementary Fig. 28**, and additional discussion has been added on page 15 in the revised manuscript.

***Comment 4:** What happens if Ketjen black carbon is used instead of nanotubes: how do performances change?*

Answer to comment 4: Woven carbon nanotube paper like buckypaper is suitable current collector for evaluating soluble organic electrode materials because of its high conductivity and the nanoscaled space holding cycled products. Ketjen black carbon was tried, but the battery capacity decays quickly. Therefore, it was not used.

***Comment 5:** Phosphodithioic acid seems very toxic and it may represent a serious danger if used in battery (as most sulfides and phosphosulfides based electrolyte as well): is it possible to replace it with a chemical less dangerous? In addition: what is its cost? It seems quite expensive for future applications.*

Answer to comment 5: We found that the organic raw material, diphenyl dithiophosphinic acid (DPDTP), can cause skin irritation and serious eye irritation. It contains no substances known to be hazardous to the environment or not degradable in waste water treatment plants from the MSDS. Some organosulfides are not friendly, but they are safe if used with caution. It is difficult to replace it with a chemical less dangerous unless the active sites are not sulfides. The chemicals used in this study are not very expensive. Wide application could lead to reduced cost for future applications.

REVIEWER 3:

***Overall Comment:** Fu and coworkers describe the synthesis of a novel organopolysulfide, 1,4-bis(diphenylphosphanyl)tetrasulfide (BDPPTS), prepared by facile electrochemical oxidation featuring the cleavage of P-S single bond and sulfur radical addition reaction. The Li/BDPPTS half-cell delivers stable cycling performance of 500 cycles with a high capacity retention of 74.8%. This paper provides a new direction to the synthesis of new organopolysulfides for cathode materials which can advance our understanding of electrochemical behaviors of organic materials in lithium-ion batteries. Further, this system exhibits an impressive discharge voltage plateau, which is beyond those of the inorganic S cathode and conventional organic electrodes. However, this work lacks the detailed rationality of the dimer (BDPPTS) design, and further explanation of their outstanding electrochemical performance needs to be addressed. Therefore, I cannot recommend publication in Nature Communications until the following issues are carefully considered and addressed:*

Answer to overall comment: We thank the reviewer for the valuable comments. The rationality of the dimer design has now been added in the Introduction section (see answer to comment 4). The outstanding electrochemical performance is also related to the π - π stacking between BDPPTS and CNTs (see answer to comment 5). Additional discussion has now been added on page 9 in the revised manuscript and additional refs. 15, 58-60 have been added.

Comment 1: Authors have done a good job including relevant references; however, there are a few more references they should consider including:

1. There are a few examples of organic electrodes system where cycling performance has improved with a limited voltage range, and that should be referenced: *Adv. Func. Mater.* 2016, 26, 6896-6903.
2. I suggest adding some references for the discharge voltage intervals for other systems including carbonyl, radical, and azo systems. These references should be included to highlight the high discharge voltage of BDPPTS system, depicted in Supplementary Figure 11.

Answer to comment 1: We thank the reviewer for the valuable comments. We have now revised the manuscript, the detailed revisions are as follows:

1. The suggested paper has been appropriately cited as **reference 18**.
2. Some references of different organic electrodes have been added to **Supplementary Fig. 11** (now Supplementary Fig. 14).

Comment 2: Authors use the terms “lithium batteries” and “Lithium-ion batteries” interchangeably. Indeed, these two terms indicate two different things: Lithium batteries are mostly non-rechargeable primary battery with lithium anode. Thus, it is more appropriate to use proper names like “rechargeable batteries” or “lithium-ion batteries” in the text.

Answer to comment 2: We agree with the reviewer. We have adopted “rechargeable batteries” and made appropriate changes in the revised manuscript including the title.

Comment 3: There is a repetition of the same sentence, which needs to be removed in Pg 8, line 139-140.

Answer to comment 3: We have deleted the repetitive sentence.

Comment 4: Authors discuss the importance of utilizing highly selective chemical methods to prepare electrode materials. They further argue that these reactions should be conducted under mild condition providing an energy-saving and “greener” option to fabricate organic electrode materials. Although these arguments can be well adopted and justify their approach to designing BDPPTS by electrooxidation reaction, they need to further address the design rationale of this dimer. They emphasize that Li/BDPPTS can discharge at higher voltage, leading to the increased power of the battery; however, there is no explanation on why such a design can lead to the increase in discharge voltage. I suspect that there could be a structural relationship such as phenyl groups (delocalizing electron density) and P atoms in the design have an impact on the redox potential. Authors should comment on why this would be the case.

Answer to comment 4: It’s a very good question. Our previous works have been focused on organosulfur with methyl or phenyl groups (*Angew. Chem. Int. Ed.* **2016**, 55, 10027-10031 and *ACS Energy Lett.* **2016**, 1, 1221-1226), which lead to discharge voltages of 2.1 – 2.2 V. For example, dimethyl disulfide shows a discharge voltage at 2.1 V. Diphenyl disulfide shows a discharge voltage at 2.2 V. When the organic group is changed to pyridyl group, dipyrindyl disulfide shows a discharge voltage at 2.45 V (*J. Mater. Chem. A* **2019**, 7, 7423-7429), meaning the electron-withdrawing groups can increase the discharge voltage. From these studies, it can be

seen that the organic groups bonded with the sulfur have profound effect on the initial discharge voltage plateaus. To further increase the discharge voltage of organosulfides, new and electron-withdrawing organic groups need to be chosen. In addition, the three-valent P is an ideal building block. Diphenyl dithiophosphinic acid contains two phenyl groups bonded with a P atom, which is bonded with sulfur. It can be determined that it has the potential to significantly increase the discharge voltage. In addition, organic tetrasulfides have decent capacities because of multi-electrons involved in the discharge reactions. Therefore, BDPPTS is an ideal structure to be synthesized. The beginning discharge voltage at 2.9 V is certainly due to the strong withdrawing groups of Ph₂P (diphenylphosphine). Additional discussions have now been added on pages 3 and 6 in the revised manuscript.

Comment 5: *It is interesting to see that BDPPTS has excellent cycling stability, although it is a small molecular system. Authors claim that the cell configuration using catholyte and CNT have been demonstrated to be useful in evaluating soluble active materials in lithium-ion batteries (In the experimental section). Is this because of intermolecular interaction such as pi-pi stacking? The dissolution in the electrolyte has been a huge hurdle for small molecular systems as organic electrodes. Authors should comment and clarify why they have chosen this specific cell configuration in more details and elaborate on this with more references with similar works on the small molecular system.*

Answer to comment 5: Catholyte can be withheld in the nanoscaled network of CNT current collector, having good conductivity and withholding capability. In addition, the molecules containing phenyl groups can have π - π stacking with CNTs, leading to stable performance. This specific cell configuration has been used in our works on soluble organic electrode materials. More references have now been added (Refs. 15, 58-60) and more discussion has been added on page 9 in the revised manuscript.

Comment 6: *One of the advantages of organopolysulfide system is that these can have higher theoretical/specific capacity ($> 500 \text{ mAh g}^{-1}$). Although Li/BDPPTS has a relatively high specific capacity (322 mAh g^{-1}), this number decreases significantly (less than 100 mAh g^{-1}) after the voltage cut-off cycling. Authors should comment on this practical capacity decrease and how does this affect on the practical application for their system.*

Answer to comment 6: It is true that BDPPTS does not have high capacity because of the high molecular weight of the organic groups which do not contribute capacities. The role of Ph₂P groups is to increase the initial discharge voltage to 2.9 V thus leading to the increasement of specific energy. To improve the practicality of BDPPTS, the cycling stability in the voltage window of 1.8-3.4 V needs to be improved, so that its high specific energy can be maintained. Strategies such as using polysulfide hosting materials or adsorption materials need to be adapted in the future. We have now added additional discussion on page 17 in the revised manuscript.

Reviewer #2 (Remarks to the Author):

I recommend to accept the paper in the present form, the authors have well answered to my questions

Reviewer #3 (Remarks to the Author):

The authors have addressed the criticisms raised by the reviewers. The paper should now be accepted.

NCOMMS-20-46262A

RESPONSE TO REVIEWERS' COMMENTS

REVIEWER #2:

Overall Comment: I recommend to accept the paper in the present form, the authors have well answered to my questions

Answer to overall comment: We thank the reviewer for the positive comment and support.

REVIEWER #3:

Overall Comment: The authors have addressed the criticisms raised by the reviewers. The paper should now be accepted.

Answer to overall comment: We thank the reviewer for the positive comment and support.